# Comprehensive Analysis for Anti-Cancer Target-Indication Prioritization of Placental Growth Factor Inhibitor (PGF) by Use of Omics and Patient Survival Data

**DOI:** 10.3390/biology12070970

**Published:** 2023-07-07

**Authors:** Nari Kim, Yousun Ko, Youngbin Shin, Jisuk Park, Amy Junghyun Lee, Kyung Won Kim, Junhee Pyo

**Affiliations:** 1Department of Medical Science, Asan Medical Institute of Convergence Science and Technology, Asan Medical Center, University of Ulsan College of Medicine, Seoul 05505, Republic of Korea; nari.kim.0908@gmail.com (N.K.); ko.yousun82@gmail.com (Y.K.); i.am.yongbin@gmail.com (Y.S.); jisukpark86@gmail.com (J.P.); amyjung.lee89@gmail.com (A.J.L.); 2Department of Radiology and Research Institute of Radiology, Asan Medical Center, University of Ulsan College of Medicine, Seoul 05505, Republic of Korea

**Keywords:** cancer metabolism, anti-cancer target, indication prioritization, drug development, *PGF*

## Abstract

**Simple Summary:**

Recognized as a promising target for anti-cancer treatment, PGF has the potential to overcome resistance to existing angiogenesis inhibitors. In this study, we aimed to identify target indications for PGF across various cancer types using bioinformatics analysis. We analyzed PGF gene function, molecular pathways, protein interactions, gene expression, mutations, survival prognosis, and tumor immune infiltration associated with PGF. The identified target diseases for PGF inhibitors included adrenocortical carcinoma, kidney cancers, liver hepatocellular carcinoma, stomach adenocarcinoma, and uveal melanoma. These findings highlight the potential of targeting PGF as a therapeutic strategy in these specific cancer types.

**Abstract:**

The expression of the placental growth factor (PGF) in cancer cells and the tumor microenvironment can contribute to the induction of angiogenesis, supporting cancer cell metabolism by ensuring an adequate blood supply. Angiogenesis is a key component of cancer metabolism as it facilitates the delivery of nutrients and oxygen to rapidly growing tumor cells. PGF is recognized as a novel target for anti-cancer treatment due to its ability to overcome resistance to existing angiogenesis inhibitors and its impact on the tumor microenvironment. We aimed to integrate bioinformatics evidence using various data sources and analytic tools for target-indication identification of the *PGF* target and prioritize the indication across various cancer types as an initial step of drug development. The data analysis included *PGF* gene function, molecular pathway, protein interaction, gene expression and mutation across cancer type, survival prognosis and tumor immune infiltration association with *PGF*. The overall evaluation was conducted given the totality of evidence, to target the *PGF* gene to treat the cancer where the *PGF* level was highly expressed in a certain tumor type with poor survival prognosis as well as possibly associated with poor tumor infiltration level. *PGF* showed a significant impact on overall survival in several cancers through univariate or multivariate survival analysis. The cancers considered as target diseases for *PGF* inhibitors, due to their potential effects on *PGF*, are adrenocortical carcinoma, kidney cancers, liver hepatocellular carcinoma, stomach adenocarcinoma, and uveal melanoma.

## 1. Introduction

In the field of cancer metabolism, there is a strong relationship between the hypoxic (low oxygen) environment and the overexpression of cancer neovascularization (formation of new blood vessels). In response to low oxygen levels, cells activate a transcription factor known as hypoxia-inducible factor (*HIF*), resulting in the overexpression of angiogenic factors.

The hypoxic environment of cells plays a critical role in the process of cancer development, proliferation, and metastasis [1]. In the early stages, hypoxia induces genetic mutations and genomic instability, leading to the transformation of normal cells into cancer cells and establishing an environment conducive to tumor growth [2]. In the cancer cell environment, *HIF* represents a transcription factor that responds to low oxygen levels, or hypoxia [3]. Activation of *HIF* in this context has significant effects. These include metabolic adaptations that promote cell survival under hypoxic conditions, increased tumor cell invasion and metastasis, suppression of the immune response against the tumor, development of resistance to certain cancer therapies, and promotion of angiogenesis [4]. Consequently, this angiogenesis process facilitates the supply of oxygen and nutrients to the tumor [3].

The oncological features of solid tumors in terms of angiogenesis include increased vessel density, abnormal vessel structure, heterogeneous blood supply, permeable vessels, and enhanced metastasis potential [5]. In addition, tumor cells release various angiogenic factors, such as vascular endothelial growth factor (*VEGF*), which stimulate the formation of new blood vessels [6]. These factors promote the proliferation and migration of endothelial cells, the building blocks of blood vessels, leading to the expansion of the tumor’s vascular network [7]. In addition, it has been reported that these factors are induced after surgery, but those levels are lower in robotic surgery, a minimally invasive surgical technique, compared to conventional surgery [8,9].

Placental growth factor (*PGF*, also known as *PlGF*) is well known as a member of the *VEGF* family, which is active in angiogenesis and endothelial cell growth, exerting an effect on its proliferation and migration [1,10]. *PGF* is a homodimeric glycoprotein, belonging to the so-called cysteine-knot superfamily of growth factors, whose members are all characterized by a common motif of eight spatially conserved cysteines, which are involved in intra- and intermolecular disulfide bonds [1]. Its mechanism of action involves binding to specific receptors, primarily *VEGFR-1*, and activating downstream signaling pathways that promote angiogenesis. *PGF* plays a crucial role in modulating endothelial cell functions, including proliferation, migration, and survival, thus contributing to the development and remodeling of blood vessels [10]. While primarily involved in physiological angiogenesis, *PGF* has also been implicated in pathological angiogenesis, such as tumor neovascularization [11]. According to Saraniti, mutations in certain genes (e.g., p53, phosphoinositide 3-kinase (PI3K)) can promote tumor growth in various cancers by upregulating PGF [12]. Understanding the genetic drivers of PGF overexpression can help identify patient subgroups that will benefit from PGF inhibition. Also, their effectiveness may be enhanced when PGF inhibitors are combined with other cancer therapies, such as radiation, chemotherapy, immunotherapy, or anti-angiogenic drugs [12]. For reference, *PGF* is commonly used to refer to the gene encoding placental growth factor, but in the context of developing anti-cancer drugs targeting this specific protein, it is more appropriate to use the term *PGF* when referring to the expressed protein [13]. The potential of PGF as a target for anti-cancer medication has been assessed. In addition, along with the significant success of bevacizumab, which is a monoclonal antibody for *VEGF-A*, *PGF* had gained emphasis as a molecule to overcome bevacizumab resistance [10]. However, from that perspective, *PGF* as a target was a half success, as aflibercept has been the only approved drug for metastatic colorectal cancer [14]. One of the reasons would be the lack of a target identification and validation process of *PGF* in various diseases.

Target identification involves finding druggable targets, such as proteins or nucleic acids, for specific diseases [15]. Successful target identification requires linking the target to the pathophysiology of the disease. In the past, targets have been identified through a series of experiments that were performed based on the hypothesis that a protein or gene may influence the pathophysiology of a disease. However, recent advances in bioinformatics and multi-omics data have led to the accumulation of big data for genetics, proteomics, and many other components. In addition, many public databases of these accumulating evidence have been opened, which have become rich sources for target identification [15].

Our research aimed to identify novel targets for anti-cancer drugs, seek to offer new possibilities for improved outcomes, and extend survival in patients facing limited or non-existent treatment options for various types of cancer. Based on the literature and preclinical data, extensive research has been conducted to explore the mechanism and role of PGF in tumorigenesis, progression, and metastasis. These investigations have pointed toward the potential of PGF as a promising therapeutic target for cancer treatment. However, only a few studies have reported the target identification process of *PGF* using comprehensive analysis based on various multi-omics databases [16]. In addition, these papers have not explored the linkage between *PGF* and the pathophysiology of specific diseases. We performed a data-based comprehensive analysis to identify potential target diseases of PGF in various solid cancers. This analysis encompassed gene expression and mutation analysis specific to different cancer types, as well as a multivariate survival analysis to assess the prognostic impact of tumor immune cells.

## 2. Methods

### 2.1. Comprehensive Analysis Framework

In our research, we followed a computational comprehensive analysis framework to assess the potential of a target molecule for cancer drug development (Figure 1). The analysis proceeded in a sequential manner, beginning with data collection from various sources, including publicly available gene expression datasets and patient survival data. Next, we performed an identification step to determine the differentially expressed genes (DEGs) associated with the target molecule. Subsequently, a functional enhancement analysis was conducted to evaluate the functional significance of the DEGs, employing enrichment analysis techniques such as gene ontology analysis and pathway analysis. Finally, we prioritized target-indications based on predetermined criteria, taking into consideration factors such as clinical relevance and therapeutic potential. By following this systematic order of analysis, we aimed to provide a comprehensive and objective assessment of the target molecule’s potential for cancer drug development.

### 2.2. Data Collection

Table 1 presents the various databases used for the comprehensive analysis to determine target-indication prioritization based on the molecular pathway and gene functions, tissue-specific distribution, PGF gene and disease association, cancer patient gene expression, and patient survival outcome.

### 2.3. Analysis of Differential Gene Expression and Mutation across Cells, Tissues, and Human Cancer Types

The gene expressions in terms of miRNA sequence, somatic mutation, and protein expression across tissues and cell types were retrieved from the web-server-based GTEx, CCLE, and Human Protein Atlas data source on 30 March 2022 using the search term “placental growth factor”. Spatial protein expression data relevant to *PGF* across human tissues and cells by immunohistochemistry and immunocytochemistry were retrieved from the Human Protein Atlas database (https://www.proteinatlas.org) (accessed on on 30 March 2022) using the search term “placental growth factor” on March 30 2022, whose source was originally from Tissue Atlas, Pathology Atlas, and Cell Atlas [19].

### 2.4. Analysis of PGF Gene Function and Molecular Pathway

Thorough gene functional information from the search term of “placental growth factor” was retrieved from the web Gene Ontology Resource database (http://geneontology.org) (accessed on on 30 March 2022). The relevant PGF molecular pathways were retrieved from the NDEx query database (https://www.ndexbio.org/iquery/) (accessed on on 30 March 2022) using the search term “placental growth factor” to determine its molecular activities in the physiological process.

### 2.5. Analysis of Target Disease Association

The Open Target Platform provides evidence of target–disease associations with an evidence score calculated from diverse data sources to aid target prioritization. The target identification and prioritization framework was based on evidence from 20 evidence sources providing target–disease relationships. The target–disease-association-score-related *PGF* was retrieved from the Open Target web-based platform (https://platform.opentargets.org) (accessed on 30 March 2022) using the search term “placental growth factor” on 30 March 2022 to investigate the evidence strength in terms of association score and scoring ranks across the diseases.

### 2.6. Univariate Survival Analysis for Prognostic Effect of PGF Expression

We performed survival analysis for evaluation of the prognostic effect of *PGF*. The GEPIA (http://gepia.cancer-pku.cn/) (accessed on 30 March 2022) provided customizable functions for RNA sequencing expression analysis according to cancer types and pathological stages, and for patient survival prognosis based on TCGA data [28].

For survival analysis, the univariate Cox proportional hazard analysis and Kaplan–Meier curves with the Log-rank test were provided to evaluate the prognostic effect of *PGF* expression on overall survival (OS) and disease-free survival (DFS) in 33 cancer types. In each cancer disease, we dichotomized the patients into high- and low-expression groups with a 50th percentile cut-off, and the survival prognostic results were retrieved on 30 March 2022 from the GEPIA solution (http://gepia.cancer-pku.cn) (accessed on 30 March 2022).

### 2.7. Multivariate Survival Analysis for Prognostic Effect of PGF Expression and Tumor Infiltrating Lymphocytes

We evaluated the impact of tumor infiltrating lymphocyte (TIL) level on survival along with PGF expression level in various cancers [29]. To determine the relationship between the differential expression of *PGF* and TIL levels of the B cell, CD8+ T cell, CD4+ T cell, macrophage, neutrophil, and dendritic cell level, we assessed the tumor IMMune Estimation Resource (TIMER) web resource (https://cistrome.shinyapps.io/timer/) (accessed on 30 March 2022).

A multivariate Cox proportional hazard model was built regarding *PGF* level and TILs including the B cell, CD8+ T cell, CD4+ T cell, macrophage, neutrophil, and dendritic cell to predict the patient survival outcome. In each cancer, the Cox proportional hazard model was fitted with the following R-code: Survival outcome (by cancer type) ~ B cell + CD8 Tcell + CD4 Tcell + Macrophage + Neutrophil + Dendritic cell + *PGF* level. The Hazard Ratios (HRs) were retrieved in each cancer.

### 2.8. CAF and MDSCs Level Correlation with PGF Expression

As important parts of the tumor microenvironment (TME), cancer-associated fibroblasts (CAFs) and myeloid-derived suppressor cells (MDSCs) actively participate in tumor metastasis by increasing the invasiveness and stemness of tumor cells and contributing to extracellular matrix (ECM) deposition [30]. We investigated the relationship between *PGF* expression levels and CAFs and MDSCs in a heat map showing the Spearman’s row correlations adjusted for purity.

We retrieved the analytic result and visualization graph of the correlation between *PGF* expression and immune infiltration level, specifically the cancer-associated fibroblast (CAF) and myeloid-derived suppressor cell (MDSC) in various cancer types from TIMER 2.0 on 30 March 2022. A heatmap of correlation value—the purity-adjusted Spearman’s rho across cancers—was provided given that tumor purity is well known as a major confounding factor.

## 3. Results

### 3.1. Differential Gene Expression and Mutation across Cells and Tissues

Across cancer type, the frequency of somatic gene mutations of PGF was rare. Except uterine corpus endometrial carcinoma (UCEC) and uterine carcinosarcoma (UCS), the somatic mutation frequency of PGF was less than 1% (Figure 2). Although PGF has little association with somatic mutation frequency, multi-layered genomic features such as copy-number variations (CNVs), methylation and ribonucleic acid (RNA) sequencing (seq) of *PGF* showed relative association across tumor types. Thus, rather than somatic mutation, the expression level would be the key information related to cancer in the case of *PGF*. The CNVs were high in adrenocortical carcinoma (ACC, 3.11), UCEC (2.62), uveal melanoma (UVM, 2.21), and pancreatic adenocarcinoma (PAAD, 2.07). In RNA-seq, expression values were high in cholangiocarcinoma (CHOL, 2.37), head-neck squamous cell carcinoma (HNSC, 2.31), UVM (3.07), and sarcoma (SARC, 2.26). This means that when a specific protein is evaluated by RNA-seq, it can be differentially expressed unlike other gene expression techniques. All TCGA cancer abbreviations are summarized in Table 2.

The DEG (Differentially Expressed Gene) analysis aims to discover quantitative changes in expression levels between the experimental groups with normalized gene expression values. DEG analysis of *PGF* in tumor cells compared to normal cells was performed across 33 cancer types with the statistical t-test. Among the tumor types, we found a statistically significant high *PGF* expression in cholangiocarcinoma (CHOL), lymphoid neoplasm diffuse large b-cell lymphoma (DLBC), glioblastoma multiforme (GBM), head and neck squamous cell carcinomas (HNSC), kidney renal clear cell carcinoma (KIRC), brain lower grade glioma (LGG), lung squamous cell carcinoma (LUSC), pancreatic adenocarcinoma (PAAD), stomach adenocarcinoma (STAD), testicular germ cell tumors (TGCT), thymoma (THYM), and uterine carcinosarcoma (UCS) (Figure 3). A low gene expression in cancer tissue vs. normal cell was found in PRAD and UCEC with statistical significance.

Regarding the effect of pathological stage on gene expression, *PGF* expression levels and pathological stages in CESC, KICH, KIRC, KIRP, and UCEC are shown in Figure 4. When we compared *PGF* expression levels in different stages of cancer tissues through pathological stage plots derived from gene expression data from GEPIA, the *PGF* gene expression levels were associated with pathological stages in CESC, KIRC, KIPR, and UCEC.

### 3.2. PGF Gene Function and Molecular Pathway

Based on the Gene Ontology database analysis, the PGF gene function is categorized into three domains: biologic process, cellular component, and molecular function. The biologic processes of PGF include the positive regulation of endothelial cell proliferation, cell–cell signaling, signal transduction, the positive regulation of mast cell chemotaxis, sprouting angiogenesis, the positive regulation of angiogenesis, and the vascular endothelial growth factor signaling pathway. The relevant PGF molecular pathways were retrieved from the NDEx query database to determine its molecular activities in the physiological process. Detailed information of gene function are presented in Appendix A.

PGF is known to bind to VEGFR1, but not VEGFR2. However, there are various ways for PGF to activate VEGFR2. If PGF expression increases, PGF/VEGF heterodimers are formed, which bind to VEGFR1/VEGFR2 receptor complexes, induce receptor cross-talk, and activate VEGFR2 [10].

In addition, there are various pleotropic actions of PGF, which are beyond the angiogenesis. First, PGF can bind to VEGFR1 on monocytes, which results in the activation of PI3K/AKT and ERK-1/2 pathways. It eventually triggers cytokine production such as TNF-α and IL-6 production and induces an inflammatory reaction and mobilization of bone-marrow-derived cells. Second, PGF can bind VEGFR1 expressed on cell surfaces of activated T-cells, which may suggest an immunomodulatory effect of PGF on T-cells. Third, PGF is able to upregulate the expression of fibroblast growth factor (FGF-2), platelet-derived growth factor (PDGF-β), and matrix metalloproteinases (MMPs), by periendothelial fibroblasts, smooth muscle cells, or inflammatory cells in tumor stroma as well. Fourth, PGF-2 and PGF-4 isoforms bind the NRP1 and NRP2, which are pleiotropic coreceptors expressed on endothelial cells as well as immune cells such as dendritic cells and regulatory T-cells (Tregs). The NRPs exert mainly inhibitory effects on immune response in the tumor.

### 3.3. Target Disease Association

In the Open Target Platform, the target–disease association score was highest in metastatic colorectal cancer (0.492), followed by non-small cell lung cancer (0.290), melanoma (0.283), and pancreatic cancer (0.130). These results are mainly derived from aflibercept, a *PGF* inhibition drug, that has been approved for metastatic colorectal cancer.

### 3.4. Univariate Survival Analysis for Prognostic Effect of PGF Expression

Among 33 cancers, 9 cancers showed a statistically significant difference in OS between the high- and low-expression group of the *PGF*. The order of prognostic effect of *PGF* for overall survival (OS) based on the hazard ratio was as follows: KICH (hazard ratio, 3.9), ACC (3.6), UVM (3.1), KIRP (2.4), DLBC (2.1), LIHC (1.8), STAD (1.5), BLCA (1.4), and LGG (0.5) (Table 3).

In the prognostic analysis on survival through the disease-free survival (DFS), 8 cancer types showed statistically significant effects with the following orders: KICH (hazard ratio, 10.0), ACC (5.9), UVM (2.4), KIRP (2.2), KIRC (1.8), BRCA (1.6), LGG (0.54), and THYM (0.33).

### 3.5. Multivariate Cox Proportional Hazards Model for Predicting Survival Outcome

Table 4 summarizes the HR with 95% confidence internals (CIs) and *p* values retrieved from Cox proportional hazard analysis in cancers, which showed a significant survival impact of PGF along with TILs: ACC, CHOL, GBM, LGG, LIHC, OV, STAD, and UCEC. These results imply that targeting PGF may affect the tumor immunity as well as patient’s survival.

### 3.6. CAF and MDSCs Level Correlation with PGF Expression

Cancer types showing a strong positive correlation between *PGF* and CAF expression were KIRP, LIHC, LUAD, PRAD, READ, STAD, and TGCT (Figure 5a). ACC, ESCA, HNSC_HPV-positive, LIHC, LUSC, OV, and TGCT cancer types showed strong positive correlations between *PGF* and MDSC expression (Figure 5b).

### 3.7. Overall Summary Evaluation

Considering the totality of the data analyses performed above, we performed an overall evaluation of which carcinomas can be target diseases of *PGF*-targeting drugs. Table 5 summarizes the dimensions of gene expression, mutations, differentially expressed genes, survival prognosis, and TME association in each cancer type. It would be reasonable to target *PGF* to treat the disease where *PGF* is highly expressed with poor survival prognosis as well as associated with high-TME components. Following the abovementioned logic, we conclude that the anti-cancer target on *PGF* would be reasonable for ACC, GBM, KICH, KIRC, KIRP, LGG, LIHC, STAD, and UVM.

## 4. Discussion

Our research demonstrated how bioinformatics tools and public databases enable us to evaluate the genetic and immune landscape of *PGF* and identify the target diseases related to *PGF*.

The functions of the PGF gene in angiogenesis have been extensively explored through thorough research, leading to a widespread understanding of its role [11,31]. Our comprehensive public-data-based analysis results are in line with the common knowledge that the *PGF* gene plays an important role in positive effect on endothelial cell proliferation, sprouting angiogenesis, and *VEGFR1* signaling. Anti-angiogenetic agents like bevacizumab hinder the delivery of oxygen and nutrients to cancer cells, ultimately inducing hypoxia [3]. In the hypoxic state, the expression of transcriptional factors such as hypoxia-inducible factor (*HIF*) is increased, which also upregulates the expression of *PGF* and its receptors such as *VEGFR1, NRP1*, and *NRP2*. This indicates that *PGF* might be an important mechanism of resistance to anti-angiogenic agent and provides an important rationale to develop *PGF* inhibitor.

In addition, our comprehensive analysis also found various pleotropic actions of *PGF* beyond angiogenesis. (1) Induction of inflammatory reaction and mobilization of bone-marrow-derived cells via activation of *PI3K/AKT* and *ERK-1/2* pathways of monocytes; (2) immunomodulatory effect of activated T-cells; (3) activation of *FGF-2*, *PDGF- β*, and *MMPs*, which can lead to desmoplasia; and (4) immune suppression via binding to NRP1 and NRP2. These pleotropic actions of *PGF* can eventually stimulate the proliferation of mesenchymal fibroblasts or cancer-associated fibroblasts (CAFs), which are closely related to the tumor microenvironment and resistance to immune checkpoint inhibitors. These results coincide with many prior reports [32,33,34,35].

The pleotropic actions of *PGF* have gained significant emphasis nowadays along with increased use of immune checkpoint inhibitors. Especially, the impact on tumor microenvironment is regarded as an important role of *PGF* contributing to tumor immunosuppression [36]. There is a lot of research to demonstrate that *PGF*-mediated actions stimulate the recruitment and activation of tumor-associated macrophages (TAMs) in the tumors [10,32,37]. It has been reported that TAMs can limit cytotoxic NK cell effector functions in the hypoxic tumor microenvironment [38]. In addition, through the *VEGFR1* pathway blockade, *PGF* can inhibit the activation and maturation of dendritic cells, which are differentiated from CD14^+^ monocytes. The lack of maturation of dendritic cells prevents the activation and proliferation of naïve CD4+ T cells and CD8+-tumor-infiltrating lymphocytes (TILs). The *NRP* receptor on T cells is involved in the Treg cells, which also induces immunosuppression [39]. These immunosuppression effects of *PGF* enhance the druggability of the *PGF* inhibitor to enhance cancer immunity, especially for combination with immune checkpoint inhibitors.

In the last part of our comprehensive analysis, the prognostic effect of *PGF* in 33 cancers was investigated based on TCGA datasets. *PGF* showed a significant impact on overall survival in several cancers through univariate or multivariate survival analysis. Throughout the survival analysis as well as all other factors, we concluded that the potential indications of the PGF-targeting drug are ACC, GBM, KICH, KIRC, KIRP, LGG, LIHC, STAD, and UVM.

So far, anti-PGF targeting strategies have gained interest for new anti-cancer drug development. There have been various drug candidates for PGF inhibition, which have mainly remained in preclinical trial stages [10]. Although many preclinical trials have demonstrated that anti-PGF drugs can inhibit tumor angiogenesis as well as tumor growth, most of them failed in clinical trials. So far, only aflibercept succeeded in FDA approval for metastatic colon cancer with the combination of the FOLFIRI regimen. Based on these results, a PGF targeting strategy might not be sufficient as a single therapy for various cancers, but it can be useful as a part of combination therapy. Indeed, bevacizumab, an anti-VEGF antibody, has been widely used as part of a combination therapy in various cancers. Similarly, interest in developing drugs that target PGF are being revisited as a part of combination therapy [10].

Especially, immune checkpoint inhibitors are combined with anti-angiogenic agents nowadays. For example, bevacizumab and atezolizumab have become the first-line chemotherapy for hepatocellular carcinoma. The mechanism of combination of anti-angiogenic agents and immune checkpoint inhibitors has not been not fully determined, but it is postulated that there are synergistic effects through modulation of the tumor microenvironment and drug delivery. For example, bevacizumab can improve the delivery of atezolizumab to the tumor site by normalizing the tumor vasculature, which can enhance the infiltration of immune cells into the tumor. This can lead to increased tumor cell killing by the activated immune cells. Moreover, bevacizumab can also decrease the levels of immunosuppressive factors in the tumor microenvironment, which can further enhance the activity of immune cells [40]. Likewise, there have been several reports that anti-PGF drugs can modulate the tumor microenvironment as well as improve tumor perfusion [33].

There are strengths and weaknesses of our comprehensive analysis. As for the strengths, it enables us to systematically collect omics data and survival data from high-quality public databases and to prioritize target indications at the early stage of drug development. Eventually, we can save time and resources by reducing in vitro and in vivo experiments. Regarding the weaknesses, the outcomes of comprehensive analysis largely rely on the quality of public open data. For example, we might not achieve goals for a rare target, if there are not enough public data to identify target information. Nevertheless, we strongly believe that comprehensive analysis based on public databases should be the starting point for target identification. These comprehensive analysis results should be validated with subsequent in vitro and in vivo experiments, which would be our future research topics.

## 5. Conclusions

We analyzed the *PGF* gene as a drug target to prioritize applicable indications through multi-dimensional comprehensive approaches. It would be reasonable to develop a new drug to target *PGF* for anti-angiogenic effect, improving tumor perfusion as well as tumor microenvironment modulation. Especially, considering the disease where *PGF* is highly expressed with poor survival prognosis as well as associated with poor tumor infiltration level, we conclude that target indications of *PGF* targeting drug would be ACC, GBM, KICH, KIRC, KIRP, LGG, LIHC, STAD, and UVM.

## Figures and Tables

**Figure 1 biology-12-00970-f001:**
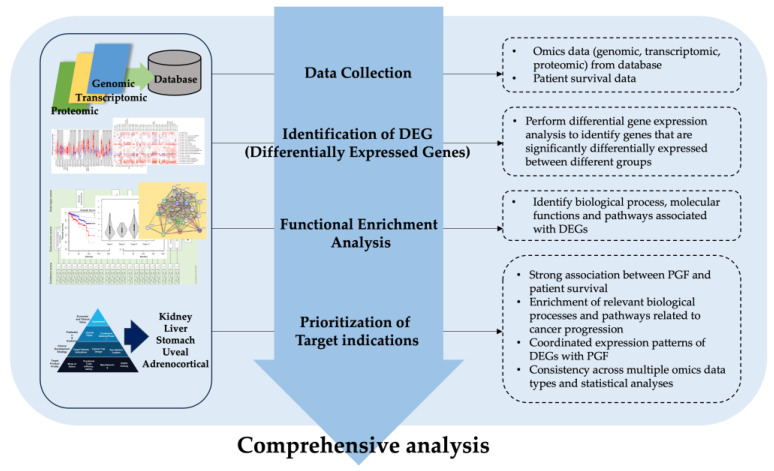
Comprehensive analysis framework.

**Figure 2 biology-12-00970-f002:**
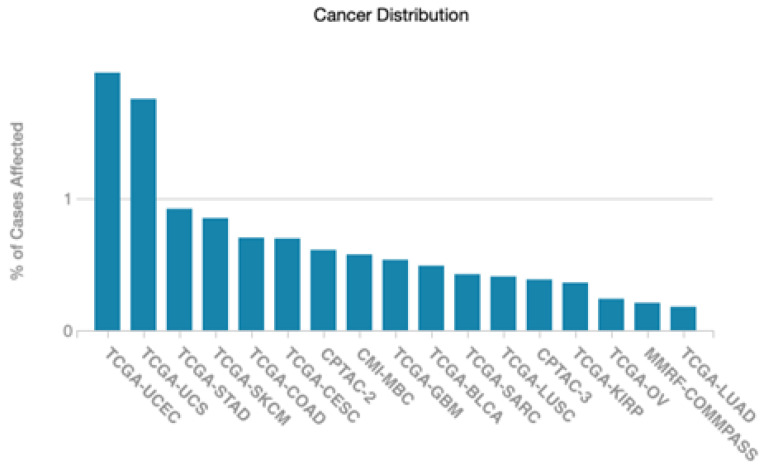
Distribution of *PGF* somatic mutations across the TCGA project cancer types. Abbreviations: TCGA, The Cancer Genome Atlas; UCEC, Uterine Corpus Endometrial Carcinoma; UCS, Uterine Carcinosarcoma; STAD, Stomach Adenocarcinoma; SKCM, Skin Cutaneous Melanoma; COAD, Colon adenocarcinoma; CESC, Cervical squamous cell carcinoma; CPTAC-2, Clinical Proteomic Tumor Analysis Consortium 2; CMI-MBC, Count Me In—The Metastatic Breast Cancer; GBM, Glioblastoma; BLCA, Urothelial Bladder Carcinoma; SARC, Sarcoma; LUSC, Lung Squamous Cell Carcinoma; CPTAC-3, Clinical Proteomic Tumor Analysis Consortium 2; KIRP, Kidney renal papillary cell carcinoma; OV, Ovarian Cancer; MMRF-COMMPASS, Multiple Myeloma Research Foundation The CoMMpass trial; LUAD, Lung Adenocarcinoma.

**Figure 3 biology-12-00970-f003:**
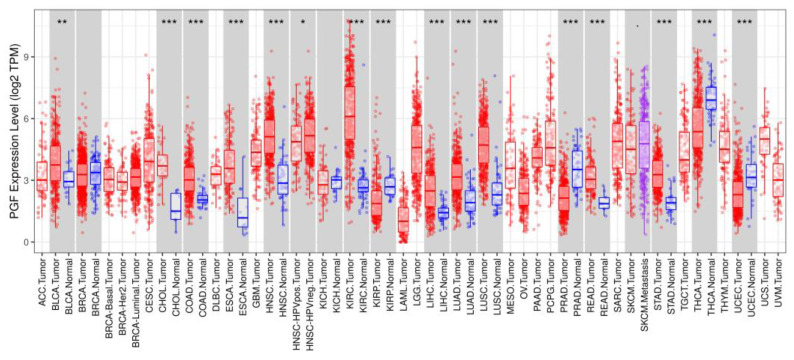
Differential *PGF* gene expression levels between tumor and adjacent normal tissue determined by DEG analysis Abbreviations: *PGF*, Placental Growth Factor; DEG, Differentially Expressed Genes; *PGF*, **: p*-value < 0.05; ***: p*-value < 0.01; ****: p*-value < 0.001.

**Figure 4 biology-12-00970-f004:**
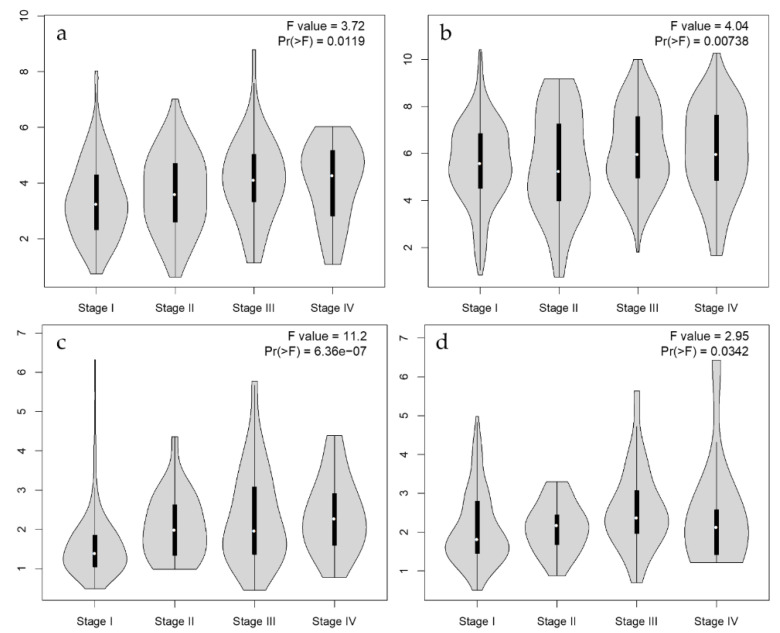
Gene expression violin plots based on patient pathological stage. In CESC (**a**), KIRC (**b**), KIPR (**c**), and UCEC (**d**), the *PGF* gene expression levels are associated with pathological stages.

**Figure 5 biology-12-00970-f005:**
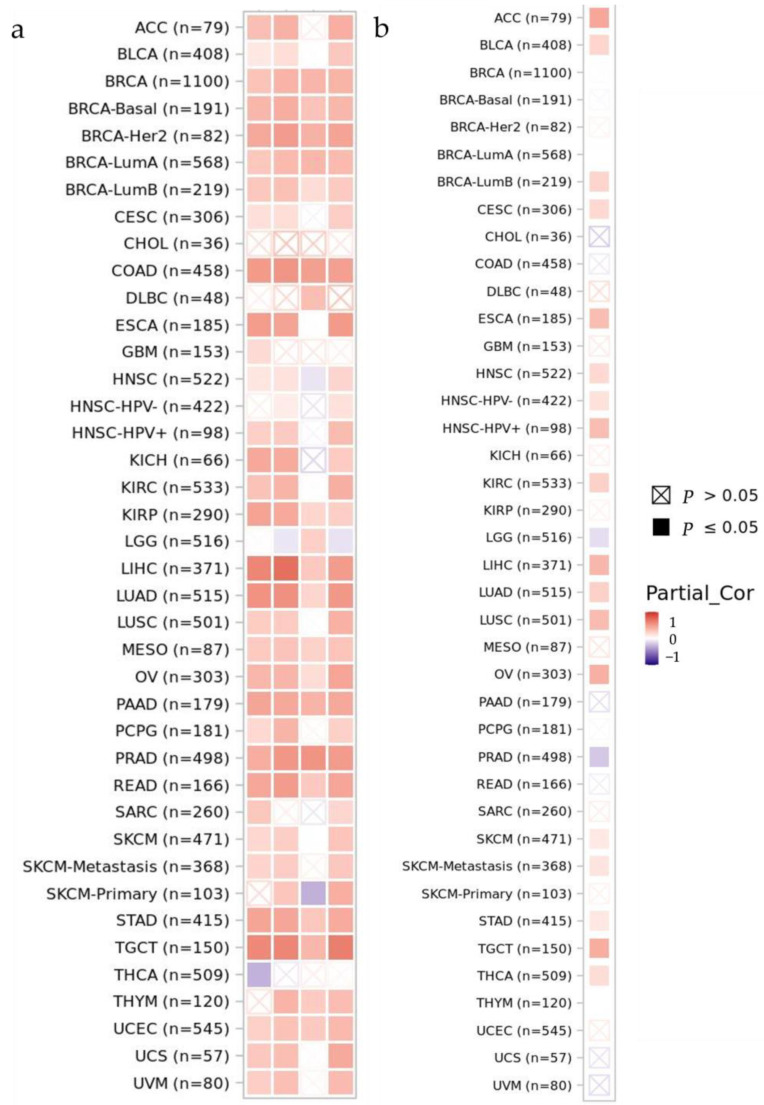
CAF and MDSCs level correlation with *PGF* expression: (**a**) CAF (**b**) MDSCs. Retrieved from the TIMER 2.0 web source on 30 March 2022.

**Table 1 biology-12-00970-t001:** Web-database for comprehensive analysis.

Database	Resources
NDEx Cytoscape Interactome Analysis Database [17]	https://www.ndexbio.org/iquery/ (accessed on 30 March 2022)
Open Target Platform [18]	https://platform.opentargets.org (accessed on 30 March 2022)
Human Protein Atlas [19]	https://www.proteinatlas.org (accessed on 30 March 2022)
GTEx (Genotype Tissue Expression) [20]	https://gtexportal.org/home/ (accessed on 30 March 2022)
Drug Gene Interaction Database [21]	https://www.dgidb.org (accessed on 30 March 2022)
Gene Ontology Resource Database [22]	http://geneontology.org (accessed on 30 March 2022)
TCGA (The Cancer Genome Atlas) Database [23]	https://www.cancer.gov/tcga (accessed on 30 March 2022)
STRING Database (Functional Protein Association Network) [24]	https://string-db.org (accessed on 30 March 2022)
CCLE (Cancer Cell Line Encyclopedia) Database [25]	https://sites.broadinstitute.org/ccle/ (accessed on 30 March 2022)
TIMER (Tumor Immune Estimation Resource) TIMER2.0 [26,27]	https://cistrome.shinyapps.io/timer (accessed on 30 March 2022) http://timer.comp-genomics.org (accessed on 30 March 2022)
GEPIA (Gene Expression Profiling Interactive Analysis) [28]	http://gepia.cancer-pku.cn (accessed on 30 March 2022)

**Table 2 biology-12-00970-t002:** TCGA cancer type abbreviation.

Abbreviation	Cancer Type
LAML	Acute Myeloid Leukemia
ACC	Adrenocortical carcinoma
BLCA	Bladder Urothelial Carcinoma
LGG	Brain Lower Grade Glioma
BRCA	Breast invasive carcinoma
CESC	Cervical squamous cell carcinoma and endocervical adenocarcinoma
CHOL	Cholangiocarcinoma
LCML	Chronic Myelogenous Leukemia
COAD	Colon adenocarcinoma
ESCA	Esophageal carcinoma
GBM	Glioblastoma multiforme
HNSC	Head and Neck squamous cell carcinoma
KICH	Kidney Chromophobe
KIRC	Kidney renal clear cell carcinoma
KIRP	Kidney renal papillary cell carcinoma
LIHC	Liver hepatocellular carcinoma
LUAD	Lung adenocarcinoma
LUSC	Lung squamous cell carcinoma
DLBC	Lymphoid Neoplasm Diffuse Large B-cell Lymphoma
MESO	Mesothelioma
MISC	Miscellaneous
OV	Ovarian serous cystadenocarcinoma
PAAD	Pancreatic adenocarcinoma
PCPG	Pheochromocytoma and Paraganglioma
READ	Rectum adenocarcinoma
PRAD	Prostate adenocarcinoma
SARC	Sarcoma
SKCM	Skin Cutaneous Melanoma
STAD	Stomach adenocarcinoma
TGCT	Testicular Germ Cell Tumors
THYM	Thymoma
THCA	Thyroid carcinoma
UCS	Uterine Carcinosarcoma
UCEC	Uterine Corpus Endometrial Carcinoma
UVM	Uveal Melanoma

**Table 3 biology-12-00970-t003:** Survival prognosis according to the DEGs of *PGF*.

Cancer Type	Outcome	HR (High vs. Low)	p(HR)	Logrank Test	No of Patients	
					High	Low
ACC	OS	3.6	0.0025	0.0013	38	38
	DFS	5.9	<0.0001	<0.0001	38	38
BLCA	OS	1.4	0.037	0.037	201	201
	DFS	1.1	0.58	0.58	201	201
BRCA	OS	1.3	0.14	0.14	535	535
	DFS	1.6	0.011	0.01	535	535
CESC	OS	1.5	0.1	0.099	146	146
	DFS	0.79	0.42	0.42	146	146
CHOL	OS	0.5	0.16	0.15	18	18
	DFS	0.7	0.44	0.43	18	18
COAD	OS	1.3	0.31	0.31	135	135
	DFS	1.0	0.9	0.91	135	135
DLBC	OS	2.1	0.38	0.37	23	23
	DFS	0.55	0.36	0.35	23	23
ESCA	OS	1	1	0.99	91	91
	DFS	1.3	0.24	0.25	91	91
GBM	OS	0.75	0.12	0.11	80	81
	DFS	0.85	0.44	0.42	80	81
HNSC	OS	1.2	0.27	0.27	259	259
	DFS	1	0.95	0.96	259	259
KICH	OS	3.9	0.092	0.069	32	32
	DFS	10	0.026	0.0058	32	32
KIRC	OS	1.3	0.088	0.086	258	258
	DFS	1.8	0.0028	0.0025	258	258
KIRP	OS	2.4	0.0068	0.0052	141	141
	DFS	2.2	0.0083	0.0068	141	141
LAML	OS	1.2	0.51	0.52	53	53
	DFS	1	na	1	53	53
LGG	OS	0.5	0.00021	0.00015	257	257
	DFS	0.54	0.00011	<0.0001	257	257
LIHC	OS	1.8	0.0012	0.00096	181	181
	DFS	1	0.82	0.82	181	181
LAUD	OS	1.1	0.41	0.41	239	239
	DFS	1.1	0.55	0.55	239	239
LUSC	OS	0.84	0.2	0.2	241	241
	DFS	1.2	0.38	0.37	241	241
MESO	OS	1.1	0.62	0.64	41	41
	DFS	1.1	0.75	0.74	41	41
OV	OS	1.2	0.24	0.23	212	212
	DFS	0.99	0.96	0.95	212	212
PADD	OS	0.89	0.59	0.6	89	89
	DFS	1.2	0.42	0.43	89	89
PCPG	OS	0.49	0.41	0.4	91	91
	DFS	0.71	0.48	0.48	91	91
PRAD	OS	0.34	0.12	0.1	246	246
	DFS	0.87	0.5	0.5	246	246
READ	OS	1.7	0.27	0.26	46	46
	DFS	1.7	0.28	0.28	46	46
SARC	OS	1.1	0.5	0.49	131	131
	DFS	0.98	0.92	0.91	131	131
SKCM	OS	0.96	0.74	0.74	229	229
	DFS	1	0.94	0.94	229	229
STAD	OS	1.5	0.011	0.0098	191	192
	DFS	1.3	0.16	0.16	191	192
TGCT	OS	2	0.58	0.58	68	68
	DFS	1.2	0.61	0.61	68	68
THCA	OS	1.3	0.6	0.6	255	255
	DFS	0.56	0.054	0.05	255	255
THYM	OS	0.74	0.68	0.67	59	59
	DFS	0.33	0.034	0.026	59	59
UCEC	OS	1.8	0.12	0.12	86	86
	DFS	0.87	0.68	0.68	86	86
UCS	OS	0.67	0.25	0.25	28	28
	DFS	0.65	0.25	0.24	28	28
UVM	OS	3.1	0.019	0.013	39	39
	DFS	2.4	0.064	0.056	39	39

Abbreviations: HR, hazard ratio; OS, overall survival; DFS, disease-free survival; DEGs, differentially expressed genes; *PGF*, placental growth factor. For cancer type abbreviations of TCGA described in Table 3, refer to Table 2. na: not available

**Table 4 biology-12-00970-t004:** Multivariate Cox proportional hazards analysis results.

Covariate	HR (95% CIs)	*p*-Value	Covariate	HR (95% CIs)	*p*-Value
Adrenocortical carcinoma (ACC)			Cholangiocarcinoma (CHOL)		
B-cell	0.00 (0.00–6.63)	0.177	B-cell	2.79 (0.00–1.01)	0.696
CD8+ T-cell	0.00 (0.00–1.63)	0.044 *	CD8+ T-cell	5.13 (132.61–1.99)	0.046 *
CD4+ T-cell	0.00 (0.00–1.14)	0.319	CD4+ T-cell	0.00 (0.00–3.01)	0.374
Macrophage	4.00 (0.00–5.76)	0.853	Macrophage	0.00 (0.00–7.11)	0.240
Neutrophil	3.23 (97,832.71–1.06)	0.029 *	Neutrophil	0.00 (0.00–1.67)	0.538
Dendritic cell	2.28 (0.00–5.23)	0.552	Dendritic cell	0.00 (0.00–3.51)	0.722
PGF expression	1.78 (1.25–2.55)	0.001 **	PGF expression	2.73 (0.10–7.06)	0.007 **
Glioblastoma Multiforme (GBM)			Low-Grade Glioma (LGG)		
B-cell	0.56 (0.32–0.97)	0.039	B-cell	0.45 (0.00–124.67)	0.784
CD8+ T-cell	1.38 (0.93–2.02)	0.101	CD8+ T-cell	214.01 (0.24–188,566.24)	0.121
CD4+ T-cell	1.08 (0.57–2.04)	0.794	CD4+ T-cell	0.113 (0.00–240.27)	0.576
Macrophage	1.16 (0.63–2.16)	0.621	Macrophage	1034.91 (26.29–40734.86)	0.000 ***
Neutrophil	1.64 (0.74–3.63)	0.221	Neutrophil	0.005 (0.00–7.62)	0.155
Dendritic cell	1.51 (1.18–1.93)	0.001 **	Dendritic cell	6.46 (0.15–274.43)	0.329
PGF expression	0.87 (0.77–0.97)	0.015 *	PGF expression	0.77 (0.67–0.88)	0.000 ***
Liver Hepatocellular Carcinoma (LIHC)			Ovarian serous cystadenocarcinoma (OV)		
B-cell	0.02 (0.00–32.78)	0.311	B-cell	0.02 (0.00–13.86)	0.255
CD8+ T-cell	0.00 (0.00–0.05)	0.002 **	CD8+ T-cell	0.04 (0.00–2.00)	0.110
CD4+ T-cell	0.00 (0.00–2.92)	0.098	CD4+ T-cell	0.00 (0.00–0.00)	0.000 ***
Macrophage	101.52 (0.35–29,455.42)	0.110	Macrophage	34,041.86 (145.80–7,947,944.83)	0.000 ***
Neutrophil	3.89 (0.00–235,124.74)	0.809	Neutrophil	2691.62 (0.45–15,895,296.64)	0.075
Dendritic cell	57.48 (1.72–1914.82)	0.024 *	Dendritic cell	0.45 (0.00–45.54)	0.735
PGF expression	0.27 (1.03–1.55)	0.021 *	PGF expression	0.73 (0.55–0.97)	0.034 *
Stomach adenocarcinoma (STAD)			Uterine Corpus Endometrial Carcinoma (UCEC)		
B-cell	87.85 (1.09–7025.32)	0.045 *	B-cell	0.32 (0.00–155.05)	0.719
CD8+ T-cell	0.26 (0.01–4.19)	0.345	CD8+ T-cell	0.00 (0.00–0.03)	0.001 **
CD4+ T-cell	0.02 (0.00–2.92)	0.126	CD4+ T-cell	0.00 (0.00–0.02)	0.002 **
Macrophage	138.70 (5.93–3244.18)	0.002 **	Macrophage	36.13 (0.25–5044.40)	0.155
Neutrophil	0.70 (0.00–153.01)	0.900	Neutrophil	5393.34 (3.75–7,742,799.81)	0.021 *
Dendritic cell	2.28 (0.18–27.87)	0.51	Dendritic cell	4.58 (0.16–126.21)	0.368
PGF expression	1.24 (1.00–1.53)	0.042 *	PGF expression	1.38 (1.15–1.65)	0.001 **

*: *p*-value < 0.05; **: *p*-value < 0.01; ***: *p*-value < 0.001.

**Table 5 biology-12-00970-t005:** Overall summary evaluation across cancer types.

	PGF Gene Expression			PGF Gene Mutation	DEG	Survival		TME	
	CNVs	Methyl	RNAseq			Uni	Multi	CAF	MDSC
ACC	***				na	***	***		**
BLCA				*	**	**			
BRCA			*			**		*	
CESC	**		*	*					
CHOL	**		**		***		***		
COAD				*	***			**	
DLBC									
ESCA					***			**	**
GBM		**		*			**		
HNSC			**		*				**
KICH			*			**		*	
KIRC					***	**			*
KIRP	**				***	**		*	
LAML		**							
LGG		***				*	***		
LIHC					***	*	**	***	**
LUAD					***			***	*
LUSC					***				**
MESO									
OV							*		
PAAD	**								**
PCPG									
PRAD	**				***			**	
READ		**			***			**	
SARC			**	*					
SKCM				**					
STAD				**	***	*	*	**	
TGCT								***	
THCA					***				**
THYM						*			
UCEC	***			***	*		***		
UCS				***		*			
UVM	***		***		na	***			

*: low correlation; **: moderate association; ***: strong association; na: not available.

## Data Availability

Not applicable.

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
