# Peer review of "Comprehensive Analysis for Anti-Cancer Target-Indication Prioritization of Placental Growth Factor Inhibitor (PGF) by Use of Omics and Patient Survival Data"

_biology, 2023, doi:10.3390/biology12070970_

Round 1

Reviewer 1 Report

  1. The first paragraph is too long and covers too many concepts. It would be better to break it into shorter paragraphs, with each focusing on one main idea, e.g. one on hypoxia and HIF, one on angiogenesis, one on PGF. This will make the introduction easier to follow.

  2. Some sentences can be made more concise by removing unnecessary words. For example, "Through the stimulation of extensive vascular networks, 39 tumors gain a growth advantage and ensure the fulfillment of their metabolic demands." can be simplified to "By stimulating vascular networks, tumors gain growth advantages and meet metabolic needs."

  3. The background on PGF can be expanded. For example, information on its molecular structure, mechanism of action, and role in physiological angiogenesis can be added. This will give readers a better understanding of PGF before its role in cancer is discussed.

  4. The rationale and objectives of the study can be made clearer. For example, a sentence on the need to identify new anti-cancer drug targets and the potential of PGF as a target can be added. The overall goal of the study to identify target cancers for PGF inhibition can also be explicitly stated.

  5.  

    Laryngeal surgery and genetic mutations are two areas that are relevant to the study of PGF in cancers. Laryngeal surgery such as partial laryngectomies is often used to treat laryngeal cancers that express high levels of PGF, suggesting it may play a role in these cancers. If PGF can be targeted as an anti-cancer treatment, it may help shrink or slow the growth of laryngeal tumors, allowing for less invasive surgeries with better preservation of laryngeal function.

    Some cancers exhibit mutations that can upregulate PGF, promoting tumor angiogenesis and growth. For example, mutations in the tumor suppressor gene p53 are common in many cancers and can activate HIF and increase PGF expression. Mutations in the phosphoinositide 3-kinase (PI3K) pathway can also increase PGF levels in cancer cells. Understanding the genetic drivers of PGF overexpression in different cancers may help identify patient subgroups that will benefit most from PGF inhibition.

    PGF inhibition may work synergistically with other anti-cancer therapies. For example, combining PGF inhibitors with radiation therapy may further starve tumors of oxygen and nutrients by reducing blood vessel growth. PGF inhibitors can also be combined with chemotherapy, immunotherapy or other anti-angiogenic drugs for increased efficacy. The effects of such combinations on tumor growth and patient survival should be investigated.

    , please discuss and cite doi:10.23812/19-282-L

  6. Some terms may need to be defined on first mention, e.g. angiogenic factors, endothelial cell, metabolites. This will ensure all readers can follow the introduction, regardless of their background.

  7. innovative and novel approach as robotic surgery represent a fascinating an promising tecniques for cancer. please discuss and cite doi:10.1002/rcs.2106

  8. The hypothesis that PGF may be a good anti-cancer target and the approaches used in this study to test this hypothesis can be highlighted in the final paragraph. This gives readers an overview of what to expect in the rest of the paper.

  9. Transitions between ideas can be improved by adding linking words like "furthermore", "moreover" and "in addition". This creates flow in the writing and shows the relationship between concepts.

  10. Parallel grammatical forms can be used to list similar ideas, e.g. "Target identification is the process to find a druggable target such as protein or nucleic acid in a specific disease [3]. The link between the target and pathophysiology of a specific disease is the key to success for target identification." can be rephrased as "Target identification involves finding druggable targets, such as proteins or nucleic acids, for specific diseases. Successful target identification requires linking the target to the pathophysiology of the disease."

minor errors

Reviewer 2 Report

Overall: The analysis and data are very valuable. The discussion section should relate to the data and be focused. The summary needs to be precise and relate to precisely what is described in the paper as submitted. 

Here are my specific comments:

Syntax and grammar

General comment: Maybe make an overall note that  that PGF is used to refer to the gene but when you refer to the expression product/ protein its better to use PlGF

 Line 17: “It is recognized”- should this perhaps say “PGF is recognized”.

Line 83: please consider changing to “whose source originally came from

Line: 163 I believe “as shown: should be changed to :are shown”

Line 164: Perhaps there should not be a paragraph break here.

Line 200: change to:  These results are mainly derived from afilibercept, a PGF inhibition drug, that has been approved for metastatic colorectal cancer.

Line 227: I think LICH should be changed to LIHC

Line 241: Please consider changing to: following the above-mentioned logic

254: Please consider changing “a common sense” to “ and become well understood

Line 258: The meaning of the following sentence is unclear; please rewrite for clarity: “As anti-angiogenetic agents such as bevacizumab in- 258 hibit supply of oxygen and nutrients to cancer cells are prohibited, leading to hypoxia 259 [21].”

Line 262: Maybe better to word as: “PGF might be an important mechanism of resistance to anti-angiogenic agents”

Content:

The discussion is problematic. As an example:

Our comprehensive analysis was able to find important protein-protein interaction 265 of PGF. PGF is known to bind to VEGFR1, but not VEGFR2. However, our analysis could 266 reveal that there are various ways for PGF to activate VEGFR2. If PGF expression in- 267 creases, PGF/VEGF heterodimers are formed, which bind to VEGFR1/VEGFR2 receptor 268 complexes, induce receptor cross-talk, and activate VEGFR2 [1]. 269 In addition, our comprehensive analysis also found various pleotropic actions of PGF 270 which are beyond the angiogenesis. First, PGF can bind to VEGFR1 on monocytes which 271 results in the activation of PI3K/AKT and ERK-1/2 pathways. It eventually triggers cyto- 272 kine production such as TNF-α and IL-6 production and induces inflammatory reaction 273 and mobilization of bone marrow-derived cells. Second, PGF can bind VEGFR1 expressed 274 on cell surfaces of activated T-cells, which may suggest immunomodulatory effect of PGF 275 on T-cells. Third, PGF is able to upregulate the expression of fibroblast growth factor (FGF- 276 2), platelet-derived growth factor (PDGF-β), and matrix metalloproteinases (MMPs), by 277 periendothelial fibroblasts, smooth muscle cells, or inflammatory cells in tumor stroma as 278 well. Fourth, PGF-2 and PGF-4 isoforms bind the NRP1 and NRP2, which are pleiotropic 279 coreceptors expressed on endothelial cell as well as immune cells such as dendritic cells 280 and regulatory T-cells (Tregs). The NRPs exert mainly inhibitory effects on immune re- 281 sponse in the tumor. These pleotropic actions of PGF can eventually stimulates the prolif- 282 eration of mesenchymal fibroblasts or cancer-associated fibroblasts (CAFs) which are 283 closely related with tumor microenvironment and resistance to immune checkpoint in- 284 hibitors. These results are coincided with many prior reports [22-25]

The reason it is problematic is because the paper is not about protein interactions or the molecular biology of VEGF. The paper is about using bioinformatics to assess which cancers could possibly benefit from treatment targeting VEGF.  The discussion in essence needs to be re-written to be more suitable to and to contextualize the data presented.

The syntax and coherence of the discussion section is unclear: it does not logically relate to the results in a clear way. The discussion section should be written clearly and with focus.

It seems to me the discussion was initially written for a larger paper that looked at gene ontology and protein interaction. 

In the conclusion section I am not sure the following approaches were used: pathway, gene ontology, comparative genomics, drug gene interaction

Overall this is a good paper and is valuable manly because of table 4 which is a GREAT CONTRIBUTION!

The discussion section should be clear and brief and explain the way the methods used arrived at the conclusion of table 4; what the strengths and weaknesses of the methods are, why the results are important and what future considerations exist.

As an aside: I think the part about TILs and TME is brilliant but the multivariate analysis using that part of the analysis did not really change the conclusion that was already formed from the expression data: that kidney tumors and ACC are a good place to start in considering a role for PlGF inhibition.

Great - in fact beautiful- up to the Discussion section. If the discussion section is re-written with focus on the data and clarity this will be a beautiful paper. My advice: keep the discussion short. The true message of the paper is table 4. 

Reviewer 3 Report

I congratulate the authors on their outstanding work. In drawing up such a protocol, we need to stick to the explanatory part, which could be divided into individual stages (below) and it would be good if you could indicate in a short table which items have been classified and which have not.

Analyzing anti-cancer target-indication prioritization requires integrating multiple data types, such as omics data (genomics, transcriptomics, proteomics) and patient survival data. In this case, the focus is on placental growth factor (PlGF) inhibition as a potential target. A comprehensive analysis framework that can be followed to prioritize target-indications would be:

Data Collection:

Obtain omics data (genomic, transcriptomic, proteomic) from cancer patients. This data should include relevant molecular profiles associated with tumor samples, such as gene expression levels, DNA mutations, and protein expression.

Collect patient survival data, including overall survival or progression-free survival, along with clinical variables such as tumor stage, grade, and treatment information.

Preprocessing and Integration:

Preprocess and normalize the omics data to ensure consistency and remove batch effects.

Integrate the different omics data types, if available, using appropriate algorithms or statistical methods to obtain a comprehensive molecular profile for each patient.

Link the omics data with patient survival data using unique patient identifiers.

Identification of Differentially Expressed Genes (DEGs):

Perform differential gene expression analysis to identify genes that are significantly differentially expressed between different groups (e.g., PlGF-high vs. PlGF-low tumors).

Apply statistical tests, such as t-tests or ANOVA, and adjust for multiple testing using methods like false discovery rate (FDR) correction.

Filter the DEGs based on fold change and statistical significance thresholds.

Functional Enrichment Analysis:

Perform functional enrichment analysis using tools such as Gene Ontology (GO) and pathway databases to identify biological processes, molecular functions, and pathways associated with the DEGs.

Assess the enrichment significance using statistical methods like hypergeometric tests or gene set enrichment analysis (GSEA).

Integration with Survival Data:

Perform survival analysis to evaluate the association between gene expression patterns (e.g., PlGF and DEGs) and patient survival.

Utilize appropriate survival analysis methods, such as Kaplan-Meier curves and Cox proportional hazards models.

Assess the significance of the association using statistical tests and adjust for confounding variables.

Prioritization of Target-Indications:

Consider the following factors for target-indication prioritization:

Strong association between PlGF and patient survival.

Enrichment of relevant biological processes and pathways related to cancer progression.

Coordinated expression patterns of DEGs with PlGF.

Consistency across multiple omics data types and statistical analyses.

Validation and Experimental Design:

Select the top-ranked target-indications for further validation in in vitro and in vivo models or clinical studies.

Design experiments to explore the functional role of PlGF and associated DEGs in cancer progression, using techniques such as gene knockdown or overexpression, cell-based assays, and animal models.

By following this comprehensive analysis framework, you can effectively prioritize target-indications for PlGF inhibition based on omics data integration and patient survival outcomes.
